# Superconducting Properties of YBa_2_Cu_3_O_7−δ_ with a Multiferroic Addition Synthesized by a Capping Agent-Aided Thermal Treatment Method

**DOI:** 10.3390/nano12223958

**Published:** 2022-11-10

**Authors:** Nur Athirah Che Dzul-Kifli, Mohd Mustafa Awang Kechik, Hussein Baqiah, Abdul Halim Shaari, Kean Pah Lim, Soo Kien Chen, Safia Izzati Abd Sukor, Muhammad Kashfi Shabdin, Muhammad Khalis Abdul Karim, Khairul Khaizi Mohd Shariff, Muralidhar Miryala

**Affiliations:** 1Laboratory of Superconductor and Thin Films, Department of Physics, Faculty of Science, Universiti Putra Malaysia (UPM), Serdang 43400, Malaysia; 2Shandong Key Laboratory of Biophysics, Institute of Biophysics, Dezhou University, No. 566 University Rd. West, Dezhou 253023, China; 3Microwave Research Institute, Universiti Teknologi MARA, Shah Alam 40450, Malaysia; 4Materials for Energy and Environmental Laboratory, Superconducting Materials, Shibaura Institute of Technology, 3 Chome-7-5 Toyosu, Koto, Tokyo 135-8548, Japan

**Keywords:** bismuth ferrite, superconductor, thermal treatment, YBCO

## Abstract

A bulk YBa_2_Cu_3_O_7−δ_ (Y-123) superconductor synthesized by a thermal treatment method was added with different weight percentages (*x* = 0.0, 0.2, 1.0, 1.5, and 2.0 wt.%) of BiFeO_3_ (BFO) nanoparticle. X-ray diffraction (XRD), alternating current susceptibility (ACS), and field emission scanning electron microscopy (FESEM) were used to determine the properties of the samples. From the XRD results, all samples showed an orthorhombic crystal structure with a *Pmmm* space group. The sample *x* = 1.0 wt.% gave the highest value of Y-123. The high amounts of BFO degraded the crystallite size of the sample, showing that the addition did not promote the grain growth of Y-123. From ACS results, the *T*_c-onset_ value was shown to be enhanced by the addition of the BFO nanoparticle, where *x* = 1.5 wt.% gave the highest *T*_c_ value (91.91 K). The sample with 1.5 wt.% showed a high value of *T*_p_ (89.15 K). The FESEM analysis showed that the average grain size of the samples decreased as BFO was introduced. However, the small grain size was expected to fill in the boundary, which would help in enhancing the grain connectivity. Overall, the addition of the BFO nanoparticles in Y-123 helped to improve the superconducting properties, mainly for *x* = 1.5 wt.%.

## 1. Introduction

Superconductivity is a phenomenon where a material exhibits zero electrical resistance and does not allow any magnetic field to penetrate through it. A material that exhibits these properties is known as a superconductor. The yttrium-barium-copper-oxide (YBCO) ceramic compound is one of the greatest discoveries for high temperature superconductors (HTSs), and it was first discovered by Paul Chu and his colleagues in 1987. YBCO is a type-II superconductor with a critical temperature (*T*_c_) of 92 K. It is the first material found to have a *T*_c_ above the boiling point of liquid nitrogen (77 K). Many applications have been explored that primarily use a superconductor, such as those in the medical and transportation fields. In the medical industry, magnetic resistance imaging (MRI) for instance, is one of its most powerful applications in diagnostic medical imaging. The superconducting magnet in MRI is a very influential tool as the magnet can achieve field values that cannot be reached by another conventional magnet. High quality, high resolution, and precise imaging can be produced as an outcome [1]. This superconductor technology has also taken place within transportation development. The Maglev Train is one of the best transportation systems in the world, where an advanced-speed train levitates above the track using superconducting magnets. This train does not depend on the friction of the track to move and stop as it travels along the guided route without any wheels. As a result, this train can reach speeds of more than 500 km/h [2]. 

In a type-II superconductor, the magnetic flux partially penetrates the bulk superconductor in the form of thin fibers called vortices. As the applied magnetic field is introduced, the loop of electric current, which is known as a screening current, is produced. The screening current interacts with the magnetic field and the external current, and this interaction moves the vortices. This vortex movement produces friction and increases energy dissipation. As more energy is dissipated, electric resistance arises, and superconductivity decreases. To reduce the dissipation of energy, the vortices need to be locked and pinned to reduce the movement. This motion of vortices can be stopped with crystal defects such as impurities and crystal lattice distortion [3]. Previous research found that additions and doping might help to produce the pinning defect in the system. This defective structure might enhance the flux pinning and result in an improvement of the critical current density *J*_c_ [4,5]. However, the *T*_c_ is also sensitive to doping and impurities such that the added element might suppress superconductivity [6]. 

In this research, a thermal treatment method is used in synthesizing the samples. This thermal treatment method, with the aid of a capping agent, has been widely used recently [7,8,9] and was chosen among other methods such as the solid-state [10], sol gel [11], and co-precipitation [12] methods because it was found that it can produce more homogeneous and finer powders, in addition to its low cost and faster and simpler step preparations [13]. The role of polyvinyl pyrrolidone (PVP) in this method, which acts as a capping agent, plays a significant role in stabilizing the particle and preventing agglomerations [8,14]. The superconductor yttrium-barium-copper-oxide (YBa_2_Cu_3_O), with the addition of a bismuth-ferrite (BFO) nanoparticle, was synthesized using the thermal treatment method. These nanoparticle materials were used in this research since they can diffuse easily during the high-temperature process. This will help to enhance the superconducting properties as they can function as strong pinning centers, unlike micro-particle materials [15,16]. The pinning centers are essential in reducing the dissipation of energy, hence improving the critical current density *J*_c_ of the superconductor [3]. Recently, the addition of magnetic impurities such as Fe_3_O_4_ and NiFe_2_O_4_ have been introduced in superconductors [17,18]. The addition of the magnetic materials is expected to increase the superconductor properties as there will be a strong interaction between the flux line network and the magnetic texture if the magnetic materials are of the same order magnitude as the flux line network. In this work, a BiFeO_3_ (BFO) nanoparticle is introduced into YBCO. BFO is a multiferroic material which exhibits both the ferromagnetic and ferroelectric order simultaneously at room temperature. It is interesting to study the effect of the addition of this magnetic material to YBCO as further studies on this material are rarely performed, especially where the thermal treatment method is used with a specific capping agent. Thus, the research was conducted to improve the properties of superconductors without significantly decreasing the *T*_c_.

## 2. Materials and Methods

YBa_2_Cu_3_O_7_ bulk ceramic was synthesized by a thermal treatment method using high purity yttrium (III) nitrate hexadrate (Y(NO_3_)_3_∙6H_2_O), barium nitrate (Ba(NO_3_)_2_), and copper (II) nitrate hemi-pentahydrate CuN_6_·2.5H_2_O as the starting materials and polyvinyl pyrrolidone (PVP) as the capping agent. All materials were purchased from Alfa Aesar chemicals (Ward Hill, MA, US). In the thermal treatment method, the starting materials and PVP were dissolved and magnetically stirred in 300 mL deionized water for 2 h at a temperature of 80 °C. The solution was then dried at 110 °C for 24 h. The solid-like green gel was obtained and then crushed and ground into a fine powder. The sample was calcined two times at 600 °C for 4 h and at 910 °C for 24 h, with intermediate grinding. Figure 1 shows the proposed mechanism of how the PVP enwrapped the ions. In this process, the carbon-carbon double bond, C=C, breaks down and interacts with the PVP monomer. The polymerization occurs and forms a giant PVP molecule. The resonance structure of the PVP attracts the corresponding ion towards its fixed position, causing a good dispersion of the particles. The PVP then enwraps the ions that existed in the solution. These ions enwrapped by the PVP have limited space for the YBCO particles to nucleate, thus decreasing the grain size of the YBCO. This limited space promotes the growth in a specific direction and orientation [14], which helps to stabilize the particles and prevent agglomerations [8]. During the calcination process, the PVP and all the organic compounds were removed and the metallic oxide was formed [13]. After the calcination process, the powder was then ground and added with different weight percentages (*x* = 0.0, 0.2, 1.0, 1.5, and 2.0 wt.%) of the BiFeO_3_ nanoparticles. The sample was continuously ground and stirred to achieve a homogenous mixture. Later, the samples were palletized with 5-ton pressure to form a bulk superconductor. From this work, four pellets with 13 mm diameters and 5 mm thicknesses were produced. Finally, the sample was sintered in a furnace with flowing O_2_ at 980 °C for 24 h and was then ready for characterization. The crystal structure and phase formation of the samples were investigated by X-ray diffraction (XRD). The microstructures of the samples were studied using a scanning electron microscope (SEM-LEO 1455 VPSEM, Zeiss, Jena, Germany). The magnetic properties of the samples were measured to determine the critical temperature, *T*_c_, using an AC susceptibility (ACS (SR830)) lock-in-amplifier at a frequency of 219 Hz with the applied field at 0.5 Oe. For this measurement, the bulk sample was cut to a bulk shape with dimensions of 5 mm × 2 mm × 1.5 mm.

## 3. Results and Discussion

### 3.1. X-ray Diffraction

Figure 2 shows the XRD pattern of the Y-123 with added BFO (*x* = 0, 0.2, 1.0, 1.5, and 2.0 wt.%). The XRD pattern shows that most of the diffraction peaks could be indexed to Y-123, which was confirmed to be the dominant phase with an orthorhombic crystal structure and the space group *Pmmm*. The Y-211 phases also appeared as a minor phase. The highest intensity diffraction pattern of Y-123 was defined at 2θ ≈ 32.67°–33.04° with miller indices of [0 1 3] and [1 0 3]. As shown in Figure 3, the intensity of [0 1 3] and [1 0 3] increased until *x* = 1.0 wt.% and started to decrease as *x* = 1.5 wt.% was added. The addition of BFO into the Y-123 system did not show any unusual changes in the polycrystalline pattern of the diffractograms unless some extra peaks were assigned, which indicated the appearance of the Y-211 phase and some impurities that belonged to the BiFeO and YBaCuFeO. The Y-211 was expected to form during the solidification of the pores in the cooling process of the Y-123 phase [8]. The volume fraction of Y-123 in the *x* = 1.0 sample gave the highest value among the samples. The purity of the Y-123 formed by using a capping agent-aided thermal treatment was quite high compared to other methods such as co-precipitation [16]. BiFeO_3_ was only observed in *x* = 0.2 in the XRD spectrum, with a small value of 0.47%. However, as a higher amount of BFO was added, the BiFeO_3_ peak could not be observed in the XRD spectrum and the YBaCuFeO_5_ started to appear. The BiFeO_3_ peak could not be seen in the other addition as the wt.% of each addition was too small [16]. The weight percentages for the phases presented are stated in Table 1.

Table 2 shows the crystallite size, lattice strain, lattice parameter, unit cell volume, and orthorhombicity of all samples. The orthorhombicity was calculated using the formula [(*b − a*)/(*a* + *b*)], where *a* is the lattice constant a and *b* is the lattice constant *b* [13]. Referring to the same table, the trend of the *b* value of the lattice parameter is aligned with the orthorhombicity value. The change in the lattice parameter could be seen as the removal of oxygen from the copper-oxygen chains occurring along the *b*-axis [19]. Meanwhile, the orthorhombicity values changed with the changes on the lattice constant due to the oxygen content in the sample [20]. As the oxygen atoms were removed, the strain energy that kept the chain directed along the *b*-axis was diminished, thus decreasing the *b* value of the lattice parameter [21].

Meanwhile, the average crystallite size can be calculated from the (103) peak using the Scherrer equation,
(1)$$D_{hkl}=\frac{k\lambda }{\beta \cos\theta },$$
where *k* is the shape factor, which is usually taken as 0.9, λ is the wavelength of the XRD CuK*α* radiation source (nm), β is the highest intensity of the main peak, and θ is the Bragg angle [22]. Referring to Figure 3, the broadening of the FWHM peak is observed as higher amounts of BFO are added. This broadening of the FWHM peak can be related to the crystallite size and lattice parameter of the samples [23]. From the figure, the broadening of the peak indicates the decrease in crystallite size, which shows that the BFO addition does not promote the crystal growth of Y-123 [24]. The lattice strain value increases as a higher amount of BFO is added, and this indicates the appearance of lattice defects such as vacancies and substitutions [23]. The addition of a high amount of magnetic material affects the symmetry of the sample [25] in which, in this sample, the appearance of the antiferromagnetic compound YBaCuFeO_5_ is believed to disturb the lattice of Y-123. 

### 3.2. Alternating Current Susceptibility (ACS) Analysis

The ACS curves consist of two steps of diamagnetic transitions which are real (χ′) and imaginary (χ″) susceptibility peaks. For the real part (χ′), the onset critical temperature, *T*_c-onset,_ and phase lock-in temperature, *T*_cj_, are observed in every sample. The *T*_c-onset_ is used to describe the transition temperature at which the sample starts to become a superconductor, where this transition occurs within the grain (intra-grain), while *T*_cj_ occurred due to the superconducting coupling between the grains (inter-grains). Meanwhile, in the imaginary part (χ″), the coupling peak temperature, *T*_p_, is observed. The *T*_p_ indicates the maximum hysteresis loss due to the motion of the intergranular vortices [26]. 

Figure 4 shows the temperature dependence of the real (χ′) and imaginary (χ″) susceptibility of Y-123 with the BFO addition (*x* = 0, 0.2, 1.0, 1.5, and 2.0 wt.%). The addition of the BFO nanoparticle enhanced the *T*_c-onset_. However, the increment of the *T*_c-onset_ value was only up to *x* = 1.5 wt.%, and then it continued to decrease as higher amounts of magnetic material were added [25]. The double-step transitions indicate the presence of the Y-211 phase [20], which happened due to the shielding of flux by the intra-grain currents as the temperature decreased [27]. The *T*_cj_ value decreased as BFO was introduced; however, the *x* = 1.5 sample gave the best values among the added samples. The high value of *T*_cj_ indicates strong superconductor inter-granular coupling between the grains [28]. The derivatives curve of *T*_cj_ plotted by Origin software is shown in Figure 5. A general view which illustrates the inter-grain current, intra-grain current, and inter-granular coupling is illustrated in Figure 6. 

As shown in the imaginary part, as the weight percentage of BFO increased, the peak shifted to the left, except for *x* = 0.2 wt.%, where its *T*_p_ increased in value for approximately 0.56 K from the pure sample. The sample *x* = 1.5 wt.% also showed a high value of *T*_p_ compared to the other added samples. A high *T*_p_ value indicates strong inter-granular solid coupling between the grains [29]. The addition of BFO helped in improving the coupling between the grains that reduced the intergranular Josephson vortices’ motion, thus improving the superconductivity [26]. The ratio of the *T*_p_/*T*_c-onset_ values in Table 3 show a declining trend where this ratio also represents the coupling between the grains. The grain coupling can be further evaluated by calculating the value of the Josephson current, *I*_o_, using the following formula:(2)$$\;I_{o}=\left(1.57\times 10^{-8}\;A/K\right)\frac{T_{c\text{-}\mathrm{onset}}{}^{2}}{T_{c\text{-}\mathrm{onset}}-T_{\mathrm{cj}}}$$
where *T*_c-onset_ is the onset critical temperature and *T*_cj_ is the phase lock-in temperature. As shown in Table 3, the *I*_o_ value increased as BFO was introduced in the Y-123, which proves that the coupling in between the grains was enhanced. The value for oxygen deficiency, δ, for all samples is also listed in the same table. These δ values are essential as the transport properties of the YBCO are strongly dependent on the δ. It is believed that the lower value of δ provided the increment of *T*_c_ on the sample [30]. Referring to Table 3, the lowest value of δ was shown by the *x* = 1.5 sample, which resulted in the highest *T*_c_ value. These values of oxygen deficiency were calculated using the formula
(3)(7 − δ) = 75.250 − 5.856c,
where δ is the oxygen deficiency and *c* is the lattice parameter’s *c*-axis [31]. The coupling temperature peak, *T*_p_, onset critical temperature, *T*_c-onset_, phase lock-in temperature, *T*_cj_, Josephson current, *I*_o_, and oxygen deficiency, δ, can be referred to as shown in Table 3.

Table 4 shows the coupling peak temperature, *T*_p_, and the intergranular critical current density, *J*_cm_, for all samples. The value of *J*_cm_ as a function of temperature was estimated using the Bean critical state model [32] with the following formula:(4)$$\;J_{\mathrm{cm}}\left(T_{p}\right)\approx \frac{H_{ac}}{\sqrt{ab}},$$
where *a* and *b* are the cross-sections of the bar-shaped sample and *H*_ac_ is the applied AC field. The *J*_cm_ value for the pure sample at 89.35 *T*_p_ was 3.88 A/cm^2^. The *J*_cm_ value decreased as 0.2 wt.% BFO was added to the sample. However, the *J*_cm_ value became higher as the BFO wt.% increased. This enhancement of the *J*_cm_ values is because of the introduction of the pinning centers in the YBCO system by the BFO [27]. The *J*_cm_ values of 1.0 wt.% and 2.0 wt.% were similar. However, the *T*_p_ value of the 1.0 wt.% was higher, which shows that the inter-grain coupling of the 1.0 wt.% samples was stronger than that of the 2.0 wt.%. The low value of *T*_p_ with a high value of *J*_cm_ may have occurred because of the flux pinning in the sample due to lattice defects such as vacancies and substitutions. The lower value of *T*_p_ shows that the flux penetration occurred in the grain boundaries at a lower temperature. The lower value of *T*_p_ also shows the weakening of the intergranular coupling between the grains [29]. In this work, *x* = 1.0 gave a higher value of *T*_p_ than *x* = 2.0, which shows that *x* = 1.0 could carry that amount of current at a higher *T*_p_. This means that the flux penetration of the grain boundary at *x* = 1.0 happened at a higher *T*_p_.

### 3.3. Field Emission Scanning Electron Microscopy (FESEM) Analysis

The morphological characterization was analyzed using the field emission scanning electron microscopy (FESEM) method. Figure 7 shows the FESEM micrographs with a 5000× magnification of the sample surface image. The average grain size of all samples was calculated from 20 randomly selected grains using ImageJ software, and they are listed in Table 5. All samples show compact microstructures with randomly distributed grains. As seen in Figure 7a, the grain size was found to be up to 20 µm; however, smaller grain sizes ranging from 0–2 µm were dominant. The average grain size for the sample was approximately 5 µm. As can be seen in Figure 7, the grain size decreased as higher wt.% amounts of BFO were added. The grain size reduction is proven with the decrease of crystallite size determined from the XRD. This reduction of grain size with the BFO addition is also observed in other methods such as the powder melt and infiltration growth (PM-IG) techniques [33] and the solid-state method [34]. However, the small grains are expected to help fill in the pores in between the grains, thus benefiting by linking the grains and enhancing the grain connectivity. These grains become tightly packed together and improve the electron transportation between the grains [13]. As could be observed in the sample *x* = 1.5 wt.%, the fine grains seemed to link up the voids between the grains. Referring to the ACS data, the *T*_p_ value for *x* = 1.5 wt.% sample was quite high, with 89.15 K, which indicates good inter-grain coupling, resulting in a high value of *T*_c_ (91.91 K) [35]. Furthermore, the fine grains also helped in increasing the vortex pinning center. Pinning may also happen with the addition of magnetic material as it may exhibit a nanophase between the YBCO grains [17]. These pinning centers can help in enhancing the *J*_c_ value of a sample. Nevertheless, as higher amounts of BFO were added, the microstructures seemed to diffuse around the sample, thus degrading the *T*_c_ value. This can be attributed to the extra addition of BFO [34]. The addition of a high magnetic moment with a lower wt.% gave a better value of *T*_c_ [25].

### 3.4. Energy Dispersive X-ray (EDX) Analysis

Energy dispersive X-ray (EDX) was used to determine the elements found in the sample. The atomic percentages of every element found are tabulated in Table 6. Based on the EDX spectra in Figure 8, the existence of the yttrium (Y), barium (Ba), copper (Cu), oxygen (O), bismuth (Bi), and ferum (Fe) peaks proved that these elements were present in the samples. However, the element of carbon (C), which is an impurity, was also detected. In the pure sample, the C peaks were apparent in the EDX spectra. Previous researchers [36] have also reported the existence of this element. These impurities of BaCO_3_ in the XRD analysis have also been reported. The suppression of BaCO_3_ might lead to another stable intermediate such as Y-211 [37]. However, in this research, BaCO_3_ was not detected in the XRD likely because the intensity of the C element was too small, with a weight percentage of only 2.78%. However, the Y-211 was still detected in the XRD (refer to Table 1). Table 6 shows the atomic percentages and atomic ratios of every element found in the samples. However, based on the table, the atomic ratio of Y:Ba:Cu did not show the 1:2:3 ratio for the added samples. This may have happened because the elemental ratio was taken randomly from the sample. 

## 4. Conclusions

The Y-123 was successfully synthesized using the thermal treatment method with PVP as the capping agent. The XRD revealed that all samples had Y-123 as the major phase and Y-211 as the secondary phase, together with some impurities such as BiFeO and YBaCuFeO. The crystallite size decreased as higher amounts of BFO were added, which shows that BFO did not promote the crystal growth of the Y-123 phase. The lattice strain came as opposition to the increment in crystallite size. The strain increased with higher percentages of BFO addition, hence showing the appearance of the lattice defects such as vacancies. From the ACS, the *T*_c-onset_ was enhanced up to 91.91 K with the BFO addition for *x* = 1.5 wt.%. However, the value then decreased as higher amounts of BFO were added. The oxygen deficiency, δ, was the lowest for *x* = 1.5 wt.%. This low value of δ gave *x* = 1.5 wt.% the highest *T*_c_ because if the δ was high, the charge carrier would become lower as the oxygen was removed at the Cu-O chain sites, thus degrading the *T*_c_ value. In terms of the microstructure properties, the FESEM revealed that the trend of the grain size was the same as that of the crystallite size in the XRD. The grain size degraded as BFO was introduced in the Y-123. Despite the degradation in the grain size, the small grains were expected to fill in the boundary and thus help in enhancing the grain connectivity. Furthermore, the fine grains may have helped in increasing the vortex pinning center and enhancing the *J*_c_. The vortex pinning also can occur with the addition of magnetic material as this material exhibits a nanophase between the YBCO grains and function as pinning centers between the grains. Overall, the addition of the BFO nanoparticles in the Y-123 helped to improve the superconducting properties, mainly for *x* = 1.5 wt.%, as higher amounts of BFO addition may degrade the quality of the superconductor.

## Figures and Tables

**Figure 1 nanomaterials-12-03958-f001:**
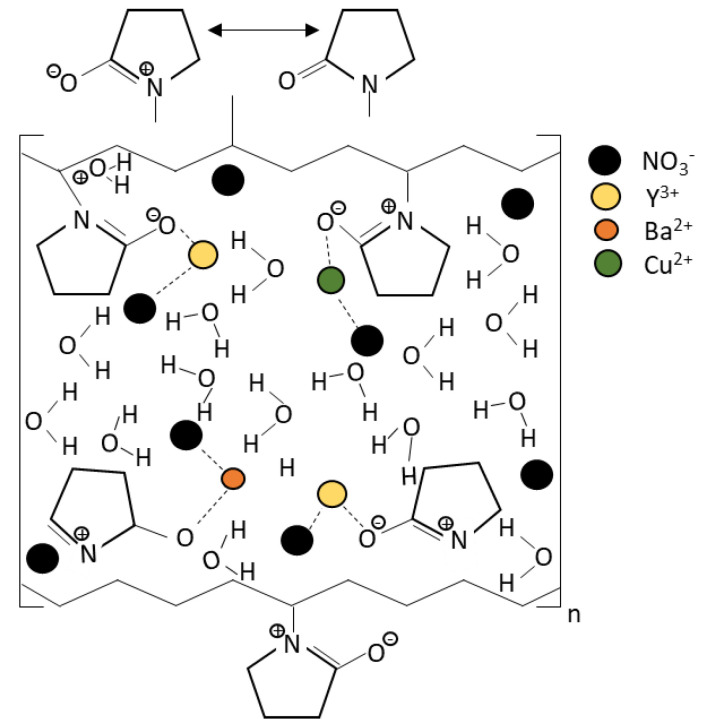
Proposed mechanism of interaction between the PVP and the metal ion [13].

**Figure 2 nanomaterials-12-03958-f002:**
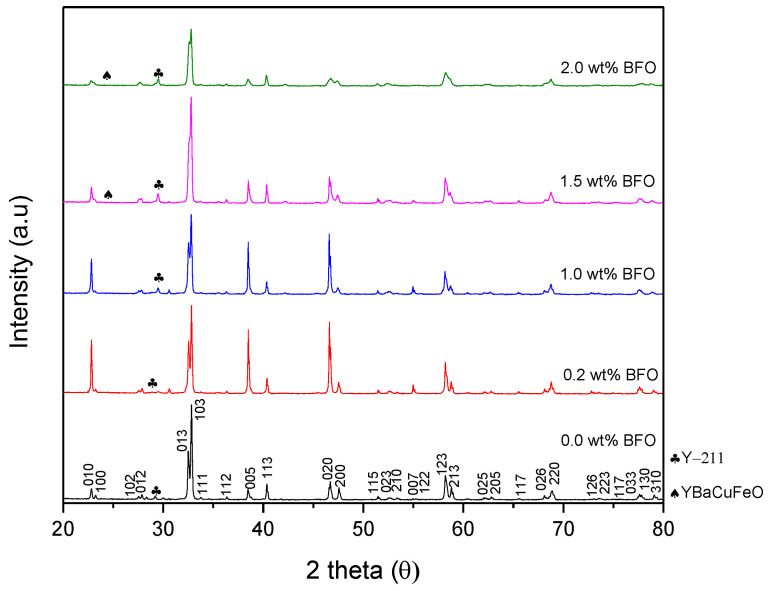
X-ray diffraction pattern of the Y-123 + *x* wt.% of BFO.

**Figure 3 nanomaterials-12-03958-f003:**
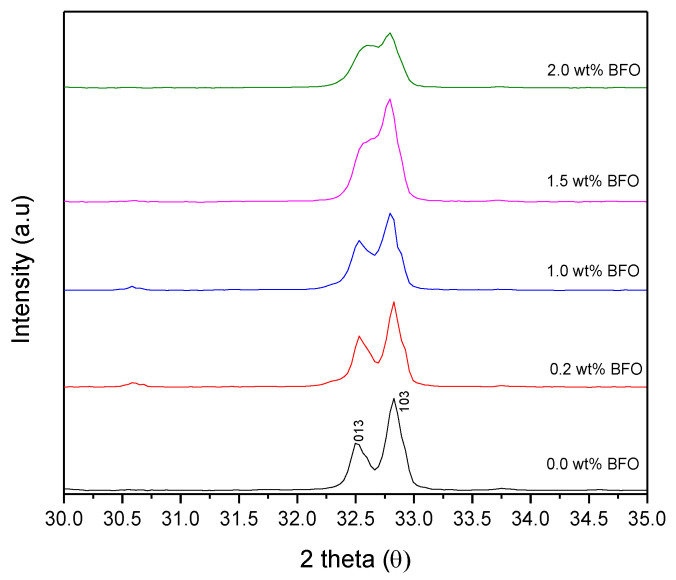
X-ray diffraction patterns corresponding to the [0 1 3] and [1 0 3] planes of the Y-123 for the Y-123 + *x* wt.% of BFO.

**Figure 4 nanomaterials-12-03958-f004:**
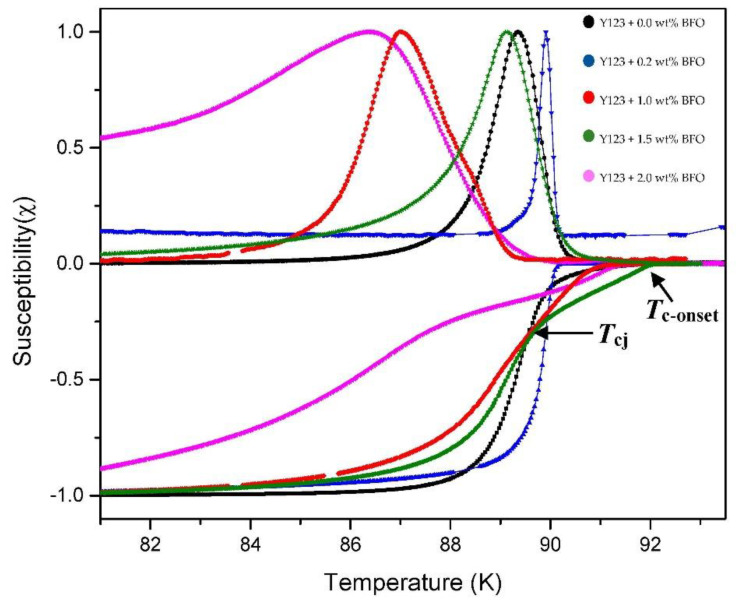
Graph of the normalized susceptibility of χ′ and χ″ against the temperature of the Y-123 + *x* wt.% of BiFeO_3_.

**Figure 5 nanomaterials-12-03958-f005:**
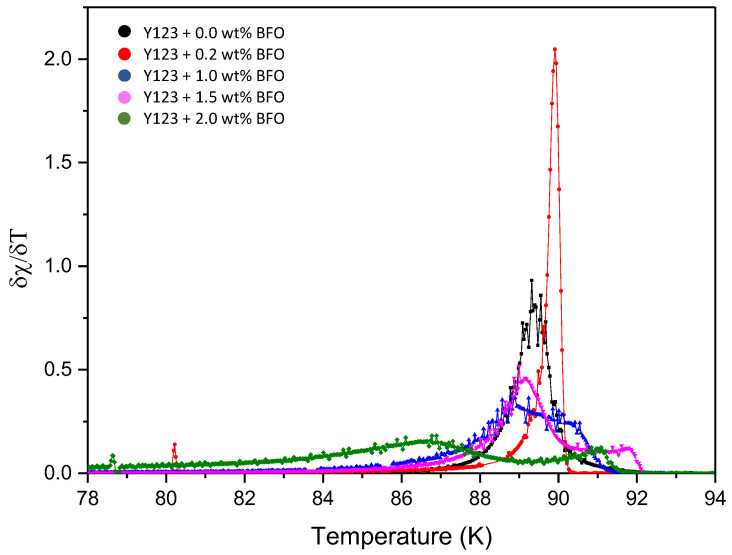
Derivatives of *T*_cj_’s susceptibility δχ/δT against the temperature of the Y-123 + *x* wt.% of BiFeO_3_.

**Figure 6 nanomaterials-12-03958-f006:**
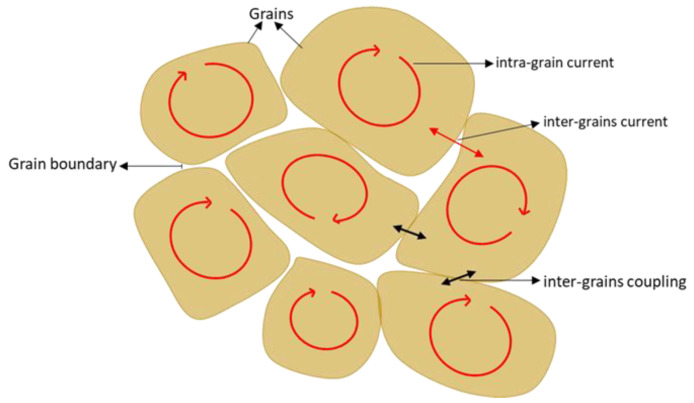
A general view of a polycrystalline sample illustrates the inter-grain current, intra-grain current, and inter-grain coupling.

**Figure 7 nanomaterials-12-03958-f007:**
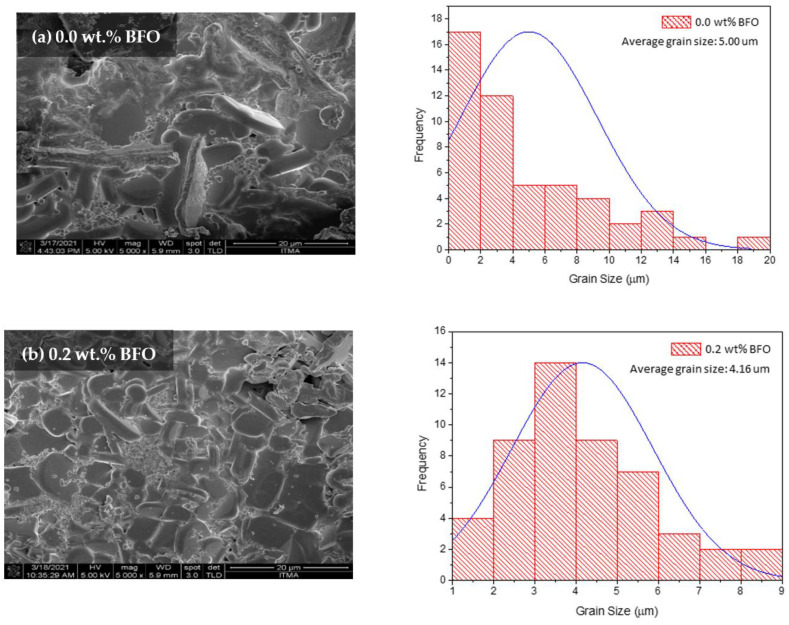
The FESEM micrographs (with a 5000× magnification) and grain size distribution for the surface Y-123 with different weight percentages of BiFeO_3_: (**a**) 0.0 wt.%, (**b**) 0.2 wt.%, (**c**) 1.0 wt.%, (**d**) 1.5 wt.%, and (**e**) 2.0 wt.%.

**Figure 8 nanomaterials-12-03958-f008:**
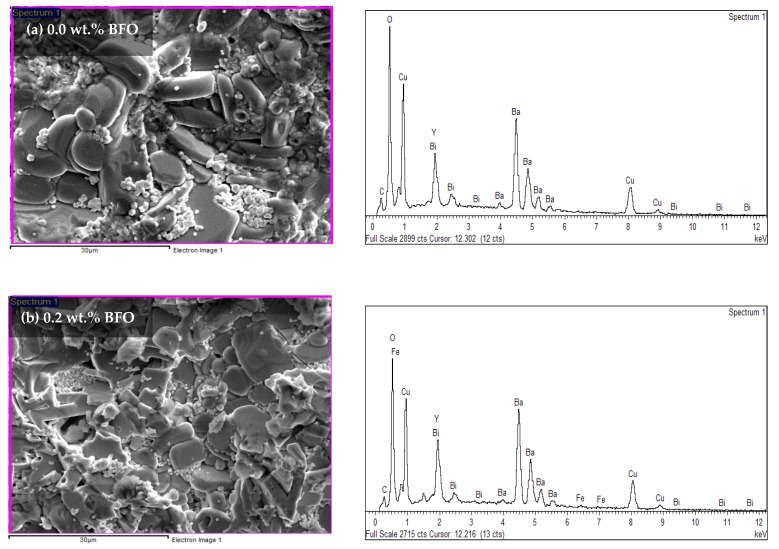
The EDX spectra for the Y-123 with different weight percentages of BiFeO_3_: (**a**) 0.0 wt.%, (**b**) 0.2 wt.%, (**c**) 1.0 wt.%, (**d**) 1.5 wt.%, and (**e**) 2.0 wt.%.

**Table 1 nanomaterials-12-03958-t001:** Weight percentages of the phases presented in the samples for the Y-123 + *x* wt.% of BiFeO_3_.

Y-123 + *x* wt.% BiFeO_3_	Weight Percentage (%)			
	YBa_2_Cu_3_O_7_	Y_2_BaCuO_5_	BiFeO_3_	YBaCuFeO_5_
0.0	90.30	9.70	0.00	0.00
0.2	94.65	4.88	0.47	0.00
1.0	97.10	2.89	0.00	0.00
1.5	92.33	4.68	0.00	2.99
2.0	88.82	6.98	0.00	4.20

**Table 2 nanomaterials-12-03958-t002:** Crystallite size, lattice strain, lattice parameters, unit cell volume, and orthorhombicity of the samples for the Y-123 + *x* wt.% of BiFeO_3_.

Y-123 + *x* wt.% BiFeO_3_	Crystallite Size (Å)	Lattice Strain (%)	Lattice Parameter			Unit Cell Volume (Å^3^)	Orthorhombicity	Full Width at Half Maximum (FWHM)
			a (Å)	b (Å)	c (Å)			
0.0	1166	0.163	3.8178 ± 0.0001	3.8837 ± 0.0001	11.6796 ± 0.0004	173.164	0.009	0.1185
0.2	1035	0.178	3.8223 ± 0.0002	3.8874 ± 0.0002	11.6770 ± 0.0003	173.507	0.008	0.1297
1.0	739	0.229	3.8281 ± 0.0002	3.8895 ± 0.0003	11.6740 ± 0.0005	173.818	0.008	0.1568
1.5	910	0.196	3.8303 ± 0.0002	3.8832 ± 0.0003	11.6717 ± 0.0006	173.603	0.007	0.1413
2.0	618	0.264	3.8385 ± 0.0005	3.8739 ± 0.0005	11.6984 ± 0.0016	173.956	0.005	0.1838

**Table 3 nanomaterials-12-03958-t003:** The coupling peak temperature, *T*_p_, onset critical temperature, *T*_c-onset_, phase lock-in temperature, *T*_cj_, Josephson current, *I*_o_, and oxygen deficiency, δ, for the Y-123 + *x* wt.% of BiFeO_3_.

Y-123 + *x* wt.% BiFeO_3_	*T*_p_ (K)	*T*_c-onset_ (K)	*T* _p/_ *T* _c-onset_	*T*_cj_ (K)	*I*_o_ (A)	Oxygen Deficiency δ
0.0	89.35	90.19	0.99	90.09	12.8 × 10^−5^	0.07
0.2	89.91	89.41	0.97	88.95	27.3 × 10^−5^	0.06
1.0	87.01	90.81	0.95	87.74	4.22 × 10^−5^	0.04
1.5	89.15	91.91	0.97	89.94	6.73 × 10^−5^	0.03
2.0	86.43	91.38	0.95	89.04	5.60 × 10^−5^	0.18

**Table 4 nanomaterials-12-03958-t004:** The cross-section area of a sample with the coupling peak temperature *T*_p_ and intergranular critical current density *J*_cm_ for the Y-123 + *x* wt.% of BiFeO_3_.

**Y-123 + *x* wt.% BiFeO_3_**	Cross-Section Area of Sample, a × b (cm^2^)	***T*_p_ (K)**	***J*_cm_ (A/cm^2^)**
0.0	0.149 × 0.296	89.35	3.88
0.2	0.170 × 0.274	89.91	3.54
1.0	0.128 × 0.196	87.01	4.17
1.5	0.151 × 0.254	89.15	3.90
2.0	0.128 × 0.196	86.43	4.17

**Table 5 nanomaterials-12-03958-t005:** The average size for the Y-123 + *x* weight % of BiFeO_3_.

Y-123 + *x* wt.% BiFeO_3_	Average Grain Size (μm)
0.0	5.00 ± 0.6
0.2	4.16 ± 0.2
1.0	4.26 ± 0.2
1.5	4.26 ± 0.4
2.0	1.97 ± 0.1

**Table 6 nanomaterials-12-03958-t006:** The atomic percentages and atomic ratios for the Y-123 + *x* wt.% of BiFeO_3_.

Y-123 + *x* wt.% BiFeO_3_	Atomic %				Atomic Ratio Y:Ba:Cu:O
	Y	Ba	Cu	O	
0.0	4.61	13.68	17.32	53.28	1:2.9:3.7:11.6
0.2	6.18	16.06	16.86	48.95	1:2.6:2.7:7.9
1.0	3.36	12.69	15.84	49.74	1:3.7:4.7:14.8
1.5	2.21	11.77	12.75	53.33	1:5.3:5.8:24.1
2.0	1.77	14.23	12.93	51.75	1:8:7.3:29.2

## Data Availability

Not applicable.

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
