# Peer review of "Superconducting Properties of YBa2Cu3O7−δ with a Multiferroic Addition Synthesized by a Capping Agent-Aided Thermal Treatment Method"

_nanomaterials, 2022, doi:10.3390/nano12223958_

Round 1

Reviewer 1 Report

Authors have described a study wherein BFO nanoparticles are added into YBCO and their properties are studied. The study is interesting, however, needs a major revision at this stage for any further consideration. 

1. The motivation to add BFO nano-particles needs its place. Its not clear why BFO! 

2. The authors need to describe the sample sizes - sample dimensions that were compacted and fabricated. What sample sizes were used for ac susceptibility measurements?

3. It appears that the prepared samples are not single grains/ single crystals. If so, how was field applied during the ac susceptibility measurements? Provide photos of the processed samples. 

4. In table-1, Why is BiFeO3 content seen only 0.2% sample? why not in other samples? What is the mechanism through which this absorption/reaction happening? Needs a systematic understanding!

5. How reproducible are these results?

6. Also, some more recent works need be cited. 

I hope the authors can revise their paper and resubmit based on these lines. Good luck!

Author Response

Thank you for your comments and susggestion. Please refer to the attachment.

Author Response

Please refer the attachement.

Round 2

Reviewer 2 Report

Authors did not address my comments properly.

The main comment: “Overall, the paper needs strong revision.” is completely ignored.

The added new papers on page 2 are not linked to the method. There is no reason monopolizing term "thermal treatment" for such very specific chemical method. Use of it in [7-9] is not a good argument either.

Added small fragments of text are full of mistakes, including “invention” of new material YBa2Cu3O (page 2), “a triangular shape of 5 mm x 2 mm x 1.5 mm dimensions” (page 3) and mysterious “FWHM peak” (page 5). This only adds more confusion to the manuscript.

Major revision of all text is necessary.

Author Response

As in the attachment
